# Is Immunotherapy Beneficial in Patients with Oncogene-Addicted Non-Small Cell Lung Cancers? A Narrative Review

**DOI:** 10.3390/cancers16030527

**Published:** 2024-01-26

**Authors:** David John McMahon, Ronan McLaughlin, Jarushka Naidoo

**Affiliations:** 1Trinity St James’s Cancer Institute, St. James’s Hospital, James’s Street, D08 NHY1 Dublin, Ireland; 2St. Vincent’s University Hospital, D04 T6F4 Dublin, Ireland; 3Department of Oncology, Sidney Kimmel Comprehensive Cancer Center, Baltimore, MD 21287, USA; 4Beaumont RCSI Cancer Centre, D09 V2NO Dublin, Ireland; 5RCSI University of Health Sciences, D02 YN77 Dublin, Ireland; 6Beaumont Hospital, D09 Y177 Dublin, Ireland

**Keywords:** immunotherapy, oncogene-addicted, oncogene-addicted non-small cell lung cancer, EGFR, ALK, KRAS, NTRK, RET, MET, BRAF, HER-2, lung cancer, TKI, tyrosine kinase inhibitors

## Abstract

**Simple Summary:**

Patients with non-small cell lung cancer (NSCLC) have a number of possible systemic treatment options, including targeted therapy, chemotherapy, immunotherapy, or antibody–drug conjugates. Approximately two thirds of lung adenocarcinomas have changes in single genes (‘oncogenes’ or oncogenic driver alterations), which drive the growth of these cancers. The role of immunotherapy in these cancers is debated, and may be different depending on the mutation present. In this review, we summarize current evidence regarding the use of immunotherapy in specific genomically driven subsets of lung adenocarcinoma. We analyze this in terms of specific mutations, focusing on both efficacy and toxicity, and potential future directions.

**Abstract:**

Over the past 20 years, there has been a paradigm shift in the care of patients with non-small cell lung cancer (NSCLC), who now have a range of systemic treatment options including targeted therapy, chemotherapy, immunotherapy (ICI), and antibody–drug conjugates (ADCs). A proportion of these cancers have single identifiable alterations in oncogenes that drive their proliferation and cancer progression, known as “oncogene-addiction”. These “driver alterations” are identified in approximately two thirds of patients with lung adenocarcinomas, via next generation sequencing or other orthogonal tests. It was noted in the early clinical development of ICIs that patients with oncogene-addicted NSCLC may have differential responses to ICI. The toxicity signal for patients with oncogene-addicted NSCLC when treated with ICIs also seemed to differ depending on the alteration present and the specific targeted agent used. Developing a greater understanding of the underlying reasons for these clinical observations has become an important area of research in NSCLC. In this review, we analyze the efficacy and safety of ICI according to specific mutations, and consider possible future directions to mitigate safety concerns and improve the outcomes for patients with oncogene-addicted NSCLC.

## 1. Introduction

Advances in our understanding of the genomic landscape and immune microenvironment of NSCLC (non-small cell lung cancer) have informed several treatment options for subsets of patients over the past 15 years. A significant proportion of patients with NSCLC, predominantly adenocarcinoma, have single identifiable mutations in cancer-causing genes (oncogenes), that drive cancer development and progression. These “driver mutations” can be found in approximately two thirds of lung adenocarcinomas and a small proportion of squamous NSCLCs, and are potential targets for targeted therapy (Figure 1 and Figure 2). Building on this, the genomic testing of tumor samples is recommended for all patients with non-squamous NSCLCs and selected squamous NSCLCs based on patients’ clinical features (never smokers, females, Asian descent), to determine the alterations present [1]. 

Therapeutically targetable oncogenes in NSCLC include mutations or alterations in Epidermal growth factor receptor mutations (*EGFRm*s), Kirsten rat sarcoma virus mutations (*KRASm*s), Anaplastic lymphoma kinase (*ALK*), B-Raf murine sarcoma viral oncogene homolog B (*BRAFm*), ROS proto-oncogene 1(*ROS1*), Mesenchymal-epithelial transition factor (*MET*), Human epidermal growth factor receptor-2 (erbB-2, or *HER2*), Rearranged during transfection (*RET*), and Neurotrophic tyrosine receptor kinase 1,2, and 3 (*NTRK*). These alterations classically appear in never smokers (e.g., *EGFR* and *ALK*), while others such as *BRAF*m and *KRAS*m can occur in patients with past smoking exposure. The presence of these alterations inform specific treatment options, mostly oral tyrosine kinase inhibitors that block the activity of the respective oncogene-driven pathways [3,4]. Recently, we have seen selected TKIs demonstrating a benefit in early-stage NSCLCs, as seen in the ADAURA (osimertinib in adjuvant *EGFR*m NSCLC) and ALINA (alectinib in adjuvant *ALK*+ NSCLC) studies [5,6].

In parallel with the progress in targeted therapy, cancer immunotherapy has become a part of the standard therapeutic strategy for advanced, and now earlier stages of, NSCLC. This is a mechanistically distinct modality of systemic therapy to targeted therapy. The most widely used class of cancer immunotherapy is immune checkpoint inhibitors (ICIs), which re-educate a patient’s adaptive immune system to induce an anti-cancer response (Figure 3). They do so by blocking the effect of ‘immune checkpoint molecules’ such as cytotoxic T-lymphocyte–associated antigen 4 (CTLA-4) (e.g., anti-CTLA4 ICIs: ipilimumab, tremelimumab), programmed cell death protein 1 (PD-1) (e.g., anti-PD-1 ICI: pembrolizumab, nivolumab, cemiplimab), lymphocyte-activation gene 3 (LAG-3) (e.g., anti-Lag3: relatlimab), or the ligand, programmed cell death ligand 1 (PD-L1) (e.g., anti-PD-L1 ICI: atezolizumab, durvalumab, avelumab). The blockade of these molecules or ligands unleashes the anti-tumor activity of T-cells. In patients with NSCLC, single agent anti-PD-1 therapy with pembrolizumab leads to a survival benefit for patients with advanced NSCLC, with a high immunohistochemical expression of PD-L1 (≥50%) on tumor cells. For those whose tumors may have any PD-L1 expression, the combination of chemotherapy plus anti-PD-1 immunotherapy, dual immune checkpoint blockade (anti-PD-1 + CTLA4), or a four-drug combination with both chemotherapy and immunotherapy agents are all treatment options [7,8,9,10].

A large variety of toxicities can occur when patients are treated with immunotherapy. These can range from mild to life-threatening and include skin, gastrointestinal, respiratory, cardiac, neurologic, endocrine, renal, and hepatic issues, amongst others, which can require complex multidisciplinary management and immunosuppressive therapies. It was noted in the early clinical development of ICIs that patients with oncogene-addicted NSCLC may have differential responses to ICI, and developing an understanding of this has become an important area of dedicated research in NSCLC. The toxicity signal for patients with oncogene-addicted NSCLC can also differ depending on the alteration present and the targeted therapy or immunotherapy used. 

Due to the increasing complexity of NSCLC, particularly adenocarcinoma, the European Society for Medical Oncology (ESMO) and the American Society of Clinical Oncology (ASCO) have begun to produce separate guidelines for oncogene-addicted and non-oncogene-addicted NSCLCs [1,11]. In these guidelines, oncogene-addicted cancers are often recommended to be treated in the first instance with either chemotherapy or targeted therapy due to superior outcomes with these systemic anti-cancer treatments, rather than immunotherapy. The optimum use of ICI in the treatment algorithm of oncogene-addicted NSCLCs is evolving over time as data emerge.

In this narrative review, we summarize the evidence to date supporting the use of ICI alone, or in combination with chemotherapy or targeted therapy, in oncogene-addicted NSCLCs. We analyze this by considering the available data on the efficacy and safety for each mutation, by considering the possible factors influencing the ICI response in these subtypes, and by considering possible future directions to mitigate safety concerns and improve outcomes for patients with oncogene-addicted NSCLCs.

## 2. Retrospective Data

Several retrospective cohorts have examined the outcomes of patients with oncogene-addicted NSCLC treated with ICIs. Firstly, the IMMUNOTARGET registry is a retrospective analysis of 551 patients with metastatic NSCLC, evaluating the response to ICI monotherapy of patients with oncogenic drivers [12]. The majority of patients (64%) were ECOG performance status 1, with a median age of 60. An ICI was given in first line in 5% of patients and ≥5th line in 14%. A phenomenon of ‘rapid progression’ (within 2 months of ICI initiation) was observed in patients whose tumors harbored *EGFR* mutations (44.8%), *ALK* rearrangements (45.5%), ROS1 fusions (42.9%), and *RET* alterations (43.8%) [12].

Retrospective studies have also been undertaken to examine the efficacy of ICIs in stage III, unresectable, NSCLCs treated with durvalumab after chemo–radiotherapy [13]. In a study by Riudevets et al., 43 patients with oncogene-addicted NSCLC were evaluated from a total of 323 patients with stage III NSCLC [13]. Of these, 19% were non-smokers. There were no statistically significant differences in median progression-free survival (mPFS, in months) between patients with known oncogenic alterations vs. those without (14.9 m (95% CI, 8.1–Not Reached [NR]) vs. 18 m (95% CI, 13.4–28.3) *p* = 1.0). In contrast, when specific oncogenic alterations were analyzed separately, the survival outcome seemed to differ depending on the oncogene involved. This analysis suggested that patients with *KRAS*m NSCLC derived a benefit from ICIs, in contrast to other reported genomic alterations. For example, mPFS was NR (11.3-NR) in the *KRAS* G12C subgroup vs. 8.1 m (5.8-NR) in *EGFR* exon 19 or 21 deletions vs. 7.8 m (7.7-NR) in the *BRAF*-mutated or *ALK*-rearranged cohort (*p* = 0.02) [13]. Similarly, another multicenter retrospective analysis of 130 patients receiving adjuvant durvalumab in stage III NSCLC after chemo–radiotherapy identified 66 patients with oncogene-addicted tumors, and again demonstrated a benefit in the *KRAS*m population: the mPFS was 12.3 months (95% CI 3.6–20.9) with and 7.2 months (95% CI 1.8–12.6) without durvalumab ([N = 41] *p* = 0.12) [14]. Four common *EGFR* mutations and twenty-one other driver alterations were identified. For those with oncogenic drivers, the mPFS was 12.3 months (95% CI 7.8–16.8) for those treated with durvalumab and 7.6 (95% CI 3.4–11.9) for patients who did not receive durvalumab (*p* = 0.038), when *EGFR*ms were excluded [14].

## 3. KRAS

Long thought ‘undruggable’, *KRAS*m NSCLC has been treated similarly to non-oncogene-addicted NSCLCs since the approval of ICIs. This consists of chemotherapy–ICI combinations, single-agent ICI, and dual ICI strategies, which are chosen based on a variety of clinical factors, the tumoral PDL-1 score, and drug access [15,16,17,18,19,20]. There is preclinical evidence that *KRAS*m NSCLC responds more favorably than other subgroups of NSCLC to ICIs [21]. This is due to the upregulation of PDL1 and high levels of CD8+ T lymphocyte infiltration [21]. The FDA and EMA have approved two *KRAS*-targeting TKIs to date, sotorasib and adagrasib- however, in contrast to other oncogene-addicted tumors, *KRAS* inhibitors are not advised to be used in the first-line setting at this time, with guidelines such as ESMO and NCCN suggesting their use in later lines of therapy [1].

### 3.1. Efficacy

In the metastatic setting, evidence of ICI efficacy in *KRAS*m NSCLC comes from prospective trials (as shown in Table 1) supported by real-word evidence—including retrospective analyses, such as the immunotarget registry (ICI monotherapy in later lines of therapy) of 246 patients discussed above, a retrospective analysis by Sun et al. (ICI monotherapy vs. chemotherapy/ICI combination) of 573 patients, and several smaller analyses [12,22,23,24,25]. Collectively, these demonstrate that there was no difference in mOS between patients with and without *KRAS* mutations for patients treated ICIs, and patients with *KRAS* mutations derive significant benefit from both chemo/ICI combinations and ICI monotherapy. 

Randomized prospective studies, such as the Keynote 042 trial of first line pembrolizumab versus chemotherapy in PD-L1 > 1% NSCLC, provide further evidence of the clinical effectiveness of ICIs in *KRAS*m patients. This demonstrated an mOS of 28 months and mPFS of 12 m in the *KRAS*m subgroup, compared to 15m and 6m, respectively, in patients without *KRAS* mutations [23]. Similar results were seen in the Keynote-189 study, with the survival benefit for chemo–immunotherapy seen in both *KRAS*m and KRASwt (wild-type) subgroups (*KRAS*m hazard ratio {HR} 0.79, 95% CI: 0.45–1.38, *KRAS*wt HR 0.55, 95%: CI 0.37–0.81) [26]. Similarly, a large Federal Drug and Safety Association (FDA) pooled analysis of 1430 patients (39% of whom had *KRAS* mutations) from 12 registrational clinical trials demonstrated no difference in outcomes between patients with *KRAS*m and *KRAS*wt tumors (Table 1) [27]. 

Statistically significant differences between specific allele subtypes are limited by small sample sizes in studies at this time. For example, the specific *KRAS* mutation did not seem to suggest a benefit in the IMMUNOTARGET registry, with G12C (*p* = 0.47) or G12D (*p* = 0.4) mutations not deriving a significantly different mPFS to those of other *KRAS* mutations [12]. This mPFS data are supported by a retrospective analysis of 137 patients with *KRAS*m NSCLC treated with chemo/immunotherapy in the first-line setting, where the mPFS was similar for both G12C and non-G12C subtypes (7.3 vs. 6.1 m [*p* = 0.12]) [25]. This patient cohort was treated at two large cancer centers, where 62 patients with G12C and 75 with other *KRAS* mutations were identified [25]. However, differences in the mOS (21 and 14 months for G12C and non-G12C [*p* = 0.24]) and ORR of (40% {25/62} vs. 31% {23/75} [*p* = 0.3]) were observed [25]. In a further retrospective review of patients receiving nivolumab in later lines of therapy, no difference in mOS was shown in the *KRAS*m vs. the *KRAS* wild-type group [21]. Another retrospective analysis of 60 patients demonstrated an mOS of 33 months in the cohort receiving immunotherapy (n = 12) and 22 months in those who did not (*p* = 0.31) [24].

In stage III NSCLCs, a retrospective analysis of patients treated with consolidation durvalumab after concurrent chemo–radiotherapy by Riudavets et al., demonstrated that patients whose tumors harbored *KRAS* mutations had the greatest benefit in terms of their mPFS (NR [11.3-NR]) of all oncogene-addicted subtypes [13].

Many oncogene-addicted NSCLC subtypes have a predilection for intracranial metastases. ICIs may have intracranial activity in *KRAS*m NSCLC. In a review of 521 patients with NSCLC and brain metastases, no difference was seen in patients with or without *KRAS* mutations, when treated with ICIs [28]. In this cohort, patients with *KRAS* mutations had improved survival if they received an ICI, although this may be confounded by lead-time bias, as many received this as a later-line therapy [28].

A growing body of evidence demonstrates that the presence of co-occurring molecular alterations in either p53, serine/threonine kinase 11(*STK11*), or Kelch-like ECH-associated protein 1 (*KEAP1*) may impact outcomes from ICIs [29]. These mutations are not direct targets of TKIs or ADCs at present but may have prognostic and predictive implications in *KRAS*m NSCLCs [30]. They may also impact the efficacy of KRAS G12C inhibitors [30]. Several studies have examined the role of co-alterations in *STK11* and *KEAP1* on outcomes from ICI monotherapy or combination therapies. Patients whose tumors harbor *KRAS/TP53* co-mutations have been associated with an improved response to ICIs in NSCLC, whereas those with *LKB1/STK11* and *KEAP1* mutations have been associated with poorer responses [21,31].

*STK11* is associated with lower levels of PD-L1 expression and is involved in cell metabolism. In patients with *KRAS*-G12C, co-mutations in *STK11* and/or *KEAP1* were associated with poorer ICI outcomes in multiple analyses, including the retrospective analysis discussed above from Dana-Farber and MSKCC, showing an mPFS of 15.8 vs. 5.1 months for those harboring these mutations (*p* = 0.01) [25]. A multicenter retrospective analysis of 174 NSCLC patients demonstrated an ORR to PD-1 blockade which differed significantly among subgroups with *STK11/LKB1* co-mutations (7.4%), *TP53* co-mutations (35.7%), and *KRAS*m patients without these alterations (28.6%) [*p* < 0.001] [32] A significantly shorter mOS was seen in the *STK11/LKB1*-mutant compared to wild-type tumors (HR 1.99, 95% CI 1.29 to 3.06; [*p* = 0.0015]) [32]. There is also prospective evidence supporting this hypothesis. In the prospective Checkmate 057 trial, the ORR to nivolumab differed significantly by co-alteration status [17]. Patients whose tumors harbored either *STK11* or *LKB1* had no response to therapy in a small subset of patients (0%, n = 0/6), while those with TP53 did respond in the majority of cases (57.1%, n = 4/7) [*p* = 0.047] [33]. It is difficult to come to a firm conclusion regarding the individual impact of these mutations, due to small numbers and confounding factors, as, in this analysis, tumors that harbored *STK11/LKB1* co-mutations also had lower levels of PDL1 expression, which may have also played a role in the poor response seen. In an abstract published describing another large retrospective analysis, the presence of *STK11* and *KEAP1* mutations in *KRAS*m NSCLCs were shown to be independent predictors of a shorter mPFS (*STK11*:HR 1.51, [*p* = 0.006]; *KEAP1*:HR 2.01, [*p* < 0.01]) and mOS (*STK11*:HR 1.81, [*p* < 0.001]; *KEAP1*:HR 2.41, [*p* < 0.0001]), which was not the case in patients without *KRAS* mutations [34]. Further analysis in prospective studies is ongoing to confirm the hypotheses that *KEAP1*, *STK11*, and *LKB1* mutations are negative prognostic factors in *KRAS*m NSCLCs, and to determine if they have a possible predictive value. Efforts to target these mutations specifically may improve survival in *KRAS*m NSCLCs, although therapeutic breakthroughs in this area are awaited.

The recent approvals of *KRAS G12C* inhibitors Adagrasib and Sotorasib for *KRAS G12C*m NSCLC provide hope for therapeutic advances for a variety of *KRAS* mutations over the coming years, and many pan-*RAS* TKIs are in development. At the present time, standard of care first-line therapy is as per the non-oncogene-addicted paradigm, with ICI, chemotherapy, or ICI–chemotherapy combinations [1]. 

### 3.2. Safety

There is a growing body of evidence suggesting potentially synergistic toxicity when treating with a combination of *KRAS G12C*-directed TKI and ICI therapy, as seen in Table 2 [21,23,35,36]. A retrospective analysis of safety data collected as part of an expanded access program for sotorasib from 102 patients with NSCLC demonstrated increased toxicity in the group of patients who received sequential sotorasib immediately after ICIs, compared to the control group who received other therapies prior to sotorasib [36].

In addition, the phase 1b/2 study CodeBreaK 100/101, combining atezolizumab or pembrolizumab with sotorasib, has demonstrated some safety concerns [37]. The toxicity of the higher-dose sotorasib arm was significant, and resulted in a reduction of sotorasib dosing due to up to 100% G3 hepatoxicity at the usual 960 mg sotorasib dose. A protocol amendment for a lead-in sotorasib monotherapy period for 21 or 42 days prior to initiation of the pembrolizumab was introduced [37]. The safety data with this approach were less concerning, and warrant further investigation. There have also been concerning features with adagrasib, to a lesser extent; however, longer follow up is needed with both sotorasib and adagrasib [38]. Also, in the retrospective analysis of NSCLC discussed above by Passiglia et al., all-grade TRAEs were significantly higher in the *KRAS*-positive group [21]. Future studies will have to proceed with caution due to these concerning safety signals with combination or sequential ICI/TKI approaches in *KRAS*m NSCLCs in light of these challenges.

**Table 1 cancers-16-00527-t001:** Selected prospective studies outlining efficacy of ICI in oncogene-addicted NSCLC.

Drug	Trial	Sample Size	PFS (Months)	OS (Months)	ORR%
** * KRAS * **					
Chemo ICI	Nakajima EC et al. [27] 2022 FDA pooled analysis	1430 *KRASm* 39% (*n* = 557) *KRASwt* 61% (*n* = 873) (*KRAS* G12C *n* = 58)	Not reported	*KRAS*m 22.4 (18.2-NR) vs. *KRAS*wt 18.7 (16.0–25.2) *KRAS* G12C 20.8 (11.3-NR) [n = 58]	*KRASm* 46% vs. *KRASwt* 47% *KRAS G12C *51%
ICI monotherapy	Nakajima EC et al. [27] 2022 FDA pooled analysis	1430 *KRASm* 39% (*n* = 557) *KRASwt* 61% (*n* = 873) *[KRAS G12C* (*n* = 45)]	Not reported	*KRAS*m 16.2 (11.1-NR) (n = 135) vs. *KRAS*wt 14.9 (12.2–6.6) (n = 322) *KRAS G12C* 11.8 (8.2-NR) (n = 45)	37% vs. 33% *KRAS G12C *33%
Pembrolizumab	Herbst RS et al. [23] 2019 Keynote 042 1st line Phase 3 PDL-1 > 50%	Any *KRAS* mutation 30 vs. *KRAS G12C* 12 vs. No *KRAS* mutation 127	Any *KRAS* mutation 12 (HR = 0.51, 0.29–0.87;95% 95% CI 8-NR) vs. *KRAS G12C* 15 (HR 0.27, 0.1–0.71; 95% CI 10-NR) vs. No *KRAS* mutation 6 (4–7) [HR 1.00 (0.75–1.34)]	Any *KRAS* mutation 29 m (HR = 0.42, 0.22–0.81; 95% CI 23-NR) vs. *KRAS G12C* NR (HR 0.28, 0.09–0.86; 95% CI 23-NR) vs. No *KRAS* mutation 15 (12–24) [HR 0.86 (0.63–1.18)]	Any *KRAS* mutation 56.7% (n = 30) (95% CI 37.4–74.5) vs. *KRAS G12C* 66.7% (n = 12) (95% CI 34.9–90.1) vs. No *KRAS* mutation 29.1 (n = 127) (21.4–37.9)
Pembrolizumab + Sotorasib Or Atezolizumab + sotorasib	Li BT et al. [37] 2022 Codebreak 100 Phase 2 1st line	58 Atezolizumab + Sotorasib lead in N = 10 Atezolizumab + Sotorasib concurrent N = 10 Pembrolizumab + Sotorasib lead in N = 19 Pembrolizumab + Sotorasib concurrent N = 19	Not reported	All patients 15.7 m (95% CI: 9.8, 17.8) Atezolizumab + Sotorasib lead in 8.1 (95% CI 2.5-NR) Atezolizumab + Sotorasib concurrent 11.5 (95% CI 5.0-NR) Pembrolizumab + Sotorasib lead in NR (95% CI 10.1-NR) Pembrolizumab + Sotorasib concurrent 14.1 (95% CI 6.2–17.8)	29% (n = 17/58)
Pembrolizumab + Adagrasib	Jänne PA et al. [38] 2022 KRYSTAL-1 (phase 1b) + KRYSTAL-7 (phase 2) 1st line	75	Not reported	Not reported	49% (n = 26/53)
** * EGFR * **					
Pembrolizumab (PDL1 > 50%)	Lisberg et al. [39] 2018 Phase 2 1st line	10	119 days	NR	0%
ABCP vs. ACP vs. BCP (A Atezolizumab B bevacizumab C Carboplatin P Paclitaxel)	Reck et al. [40] 2019 Impower150 1st line Phase 3	122 ABCP (34) vs. ACP (45) vs. BCP (43)	ABCP 10.2 vs. BCP 6.9 (HR 0.61) Sensitizing mutations ABCP vs. BCP (HR 0.41)	ABCP not estimable (NE) vs. 18.7 BCP Sensitizing mutations ABCP NE vs. 17.5 BCP (HR 0.31)	ABCP (70.6%) Vs. ACP (35.6%) Vs. BCP (41.9%)
Durvalumab + Gefitinib	Gibbons D et al. [41] 2016 Phase 1 1st line	19 9 ARM A (concurrent) 10 ARM B (sequential)	Not reported	Not reported	ARM A 77.8% (n = 7) Arm B 80.0% (n = 8)
Nivolumab +Erlotinib	Gettinger S et al. [42] 2018 Phase 1 ≥1st line (prior chemotherapy excluded)	21	5.1 (95% CI: 2.3–12.1)	18.7 (95% CI: 7.3–NR)	15% (n = 3 of 20)
Sintilimab + IBI305 + chemotherapy or Sintilimab + chemotherapy vs. chemotherapy alone.	Lu S et al. et al. [43] 2023 Orient-31 trial Phase 3 2nd line	476	Sintilimab, (anti-PD1) + chemotherapy 5.5 vs. Chemotherapy alone 4.3 Sintilimab + IBI305 + chemotherapy 7.2 vs. 4.3 Sintilimab + chemotherapy versus 19.2 Chemotherapy alone	Sintilimab + IBI305 + chemotherapy 21.1 vs. Sintilimab plus chemotherapy 20.5 vs. Chemotherapy alone 19.2	35% (n = 55/158) sintilimab + chemotherapy vs. 29% (n = 47/160) chemotherapy alone
Durvalumab + Osimertinib	Oxnard g et al. [44] 2022 TATTON trial Phase 1b 2nd line	23	Not reported	Not reported	43% (approximate)
Nivolumab	Rizvi et al. [45] 2014 Checkmate 063 Phase 2 ≥2nd line	117	1.9 (95% CI 1.8–3.2)	8.2 (95% CI 6.1–10.9)	14.5% (95% CI 8.7–22.2)
Durvalumab + Osimertinib vs. Osimertinib alone	Yang J et al. [46] 2019 CAURAL trial Phase 3 ≥2nd line ** Terminated early due to the TATTON trial safety concerns*	14	Combination NR vs. 19.3 Osimertinib	NR vs. NR * early termination	Combination 80% (n = 12 of 15) ([95% CI: 52–96]) vs. Osimertinib 64% (n = 9 of 14) ([95% CI: 35–87]).
Nivolumab	Mok T et al. [47] 2022 Checkmate 722 ≥2nd line Phase 3	294	Nivolumab + chemotherapy 5.6 vs. chemotherapy 5.4 (HR0.75, *p* = 0.0528)	19.4 vs. 15.9 (HR 0.82, *p* ≥ 0.05)	31% vs. 27% ** Exact number at risk unknown*
Pembrolizumab	Chih-Hsin Yang J et al. [48] 2023 KEYNOTE 789 ≥2nd line Phase 3	492	Pembrolizumab + chemotherapy 5.6 vs. chemotherapy 5.5 (HR 0.80; *p* = 0.0122)	15.9 vs. 14.7 m (HR 0.84; *p* = 0.0362)	29.0% vs. 27.1%
** * EGFR* + *ALK* + combined **					
Durvalumab * *EGFR* and *ALK groups combined f*or *analysis*	Naidoo et al. [35] 2022 PACIFIC subgroup analysis Phase 3 -Stage III	35 24 (durvalumab) vs. 11 (placebo) (Median follow up 42 months)	11.2 vs. 10.9 (95% CI 7.3–20.7 vs. 1.9-NR, [HR 0.91]	46.8 (95% CI 29.9-NR) vs. 43.0 (95% CI 14.9, NE)	26% (95% CI, 10.2, 48.4) vs. 18.2% (95% CI 2.3, 51.8)
Durvalumab * (*EGFR* +/*ALK*+ combined)	Garassino MC et al. [49] 2018 ATLANTIC Phase 2 ≥3rd line	444 111 in cohort 1 (*EGFR*+/*ALK*+)	PDL-1 ≤ 25% 1.9 (1.8–1.9) PDL-1 ≥ 25% 1.9 (1.8–3.6)	PDL-1 ≤ 25% 9.9 (4.2–13.0) PDL-1 ≥ 25% 13.3 (8.1-NR)	14% [12.2% (n = 9 of 74) of PDL1 + ≥25% group 95% CI 5.7–21.8)]
** * ALK * **					
Nivolumab + Crizotinib	Spigel D et al. [50] 2018 CheckMate 370 Phase 1/2 1st line	13	Not reported	Not reported	38% (n = 5 of 13)
Pembrolizumab + Crizotinib	Patel SP et al. [51] 2020 Phase 1b 1st line	Not reported	Not reported	Not reported	Not reported
Atezolizumab + Alectinib	Kim DW et al. [52] 2022 Phase 1b 1st line	21	NR (95% CI: 13–NR)	NR (95% CI: 33 –NR)	86% (n = 18)
Avelumab + Lorlatinib	Shaw AT et al. [53] Javelin Lung 01 2018 Phase 1b ≥2nd line	28	Not reported	Not reported	46.4%
Nivolumab + Ceritinib	Felip E et al. [2] 2017 Phase 1 1st or ≥2nd line	36	Not reported	Not reported	63% (pretreated) 83% (TKI Naïve)
Durvalumab -	Naidoo et al. [35] 2022 PACIFIC subgroup analysis Phase 3 Stage III	4	7.8 [95% CI, 3.9-NR]	Not reported	Not reported

## 4. EGFR 

### 4.1. Efficacy

There have been both prospective and retrospective studies analyzing the utility of ICIs in patients with classical activating *EGFR*-mutant NSCLC. In stage III disease, small prospective and retrospective studies have demonstrated an ORR of 26%, mPFS of 8.1–11.2 m, and mOS of 46.8 months with durvalumab maintenance after definitive chemo–radiotherapy (Table 1) [12,13,34,35,54]. In the overall population of the PACIFIC trial of durvalumab consolidation after chemo–radiotherapy, a similar ORR (30%) and mOS (47.5 months) were demonstrated, albeit with a longer mPFS (16.9 months) [54]. 

In the metastatic setting, evidence supporting a single-agent ICI in *EGFR*m NSCLC is lacking, even among those with a PD-L1 expression >50%. In the second-line Keynote 010 study, a significantly shorter mOS was seen in *EGFR*m patients than in *EGFR*-wt patients when both were treated with pembrolizumab (6.5 vs. 15.7 months) [55].

Evidence for chemotherapy–ICI combinations is conflicting. There is a suggestion of synergistic efficacy with the addition of *VEGF* inhibition [40,56]. This is mainly based on the Impower150 trial, which demonstrated an improved mOS and mPFS with bevacizumab + chemotherapy + ICI versus bevacizumab + chemotherapy alone in patients with sensitizing *EGFR* mutations [40]. Similarly, the phase 3, second-line ORIENT-31 trial showed an mPFS of 5.5 months vs. 4.3 months (HR 0.72 [95% CI 0.55–0.94]; two-sided *p* = 0.016) for ICIs (sintilimab, anti-PD1) + chemotherapy vs. chemotherapy alone [43]. A third arm of sintilimab plus IBI305 (a VEGF inhibitor) plus chemotherapy compared with chemotherapy alone demonstrated an mPFS of 7.2 months [95% CI 6.6–9.3]; HR: 0.51 [0.39–0.67]; two-sided *p* < 0.0001). With a median follow up of 12.9–15.1 months, the mOS was 21.1 months (95% CI 17.5–23.9) for sintilimab plus IBI305 plus chemotherapy (HR 0.98 [0.72–1.34]) and 20.5 months (15.8–25.3) for the sintilimab plus chemotherapy group (HR 0.97 [0.71–1.32]) versus 19.2 months (15.8–22.4) for chemotherapy alone [43]. While extrapolating the data from this trial is difficult, as no VEGF inhibition was given in the control arm, it is nonetheless a positive phase 3 trial of ICI + chemotherapy in the second-line setting for *EGFR*m NSCLCs.

These trials conflict with the majority of evidence evaluating the use of ICIs in *EGFR*m patients, showing no benefit to the addition of an ICI in metastatic *EGFR*m NSCLC, for example, the second-line Checkmate 722 study, where no benefit was seen with the addition of nivolumab to chemotherapy after *EGFR*-directed TKI, and a metanalysis investigating ICI use vs. docetaxel in second-line trials (Keynote 010, OAK, POPLAR, Checkmate 017 and Checkmate 057) (Table 1) [47,57]. In this metanalysis, the *EGFR*-mutant subgroup of 186 patients had a pooled HR for an mOS of 1.05 (95% CI: 0.70–1.55, *p* < 0.81; heterogeneity *p* = 0.80). Also, the mOS did not differ significantly according to the PD-L1 expression in the *EGFR*-mutant patients (TPS > 50% 6.5 months vs. TPS < 1% 5.7 months). Similarly, the recently presented KEYNOTE 789 trial of 492 patients randomized to chemotherapy with pembrolizumab or placebo in TKI-resistant *EGFR*m NSCLC failed to demonstrate a significant improvement in mPFS (5.6 vs. 5.5 m HR 0.80; *p* = 0.0122) or mOS (15.9 vs. 14.7 m HR 0.84; *p* = 0.0362) after a median follow up of 42 months [48]. In the third-line or greater setting, the ATLANTIC study demonstrated an ORR of 14% for *EGFR*-mutant patients treated with durvalumab [49]. Early phase trials combining ICIs and *EGFR* TKI have yielded an ORR varying from 0–79% [39,41,44,45].

In the IMMUNOTARGET registry, different responses and mPFSs were noted between a variety of *EGFR* mutations. The shortest mPFS was 1.4 months in the *T790M* and complex mutations subgroup, improving to 1.8 months for exon 19, 2.5 months for exon 21, and 2.8 months for other mutations (*p* = 0.001) [12]. Further detail regarding prospective and retrospective evidence for the use of ICIs in *EGFR*m NSCLC is seen in Table 1 and Table 2. 

**Table 2 cancers-16-00527-t002:** Summary table of selected prospective data examining the safety of immunotherapy in oncogene-addicted NSCLCs.

Drug	Trial	Sample Size	TRAE	IRAE
** * KRAS * **				
Pembrolizumab + Sotorasib or Atezolizumab + Sotorasib	Li BT et al. [37] 2022 Codebreak 100 Phase 2 1st line	58 Atezolizumab + Sotorasib lead In N = 10 Atezolizumab + Sotorasib concurrent N = 10 Pembrolizumab + Sotorasib lead in N = 19 Pembrolizumab + Sotorasib concurrent N = 19	All grade 88% (n = 51) G3/4 59% (n = 34)	AE of special interest Hepatotoxicity G3/4 43% (n = 25)
Pembrolizumab + Adagrasib	Jänne PA et al. [38] 2022 KRYSTAL-1 (phase 1b) + KRYSTAL-7 (phase 2) 1st line	75	All grade 83% G3/4.44% G3 Elevated lipase 11% G3 increased ALT/AST 8%/9%	Not reported
** * EGFR * **				
Pembrolizumab	Lisberg et al. [39] 2018 Phase 2 1st line PDL1 > 50%	10	46%	46%
ABCP vs. ACP vs. BCP (A, Atezolizumab B, bevacizumab C, Carboplatin P, Paclitaxel)	Reck et al. [40] 2019 Impower150 Phase 3 1st line	122 ABCP (34) vs. ACP (45) vs. BCP (43)	G3/4 ABCP 64% (n = 21 of 33) vs. ACP 68% (n = 30 of 44) vs. BCP 64% (n = 28 of 44) **+ 1 G5 toxicity**	ABCP 55% (n = 18) G3/4 9% (n = 3) vs. ACP 52% (n = 23), G3/4 9% (n = 4) vs. BCP 23% (n = 10), G3/4 2% (n = 1) *AE of special interest including irAE
Durvalumab + Gefitinib	Gibbons D et al. [41] 2016 Phase 1 1st line	19 9 ARM A (Concurrent) 10 ARM B (Sequential)	All grade AE 100%	Not reported
Nivolumab + Erlotinib	Gettinger S et al. [42] 2018 Phase 1 ≥1st line (prior chemotherapy excluded)	21	All grade 100% (n = 21) rash (n = 10, 48%) fatigue (n = 6, 29%) paronychia (n = 6, 29%), skin fissures (n = 5, 24%) No G4 or 5 toxicities	All grade N = 18 (86%) 24% (n = 5) ≥G3 toxicities Diarrhea (n = 2), ALT+/− AST increase (n = 2) Weight loss (n = 1)
Sintilimab + IBI305 + chemotherapy or Sintilimab + chemotherapy vs. chemotherapy alone	Lu S et al. [43] 2023 Orient-31 trial Phase 3 2nd line	476	≥3 56% (n = 88/158) sintilimab + IBI305 + chemotherapy group vs. 41% (n = 64/156) in the sintilimab + chemotherapy group vs. 49% (n = 79/160) in the chemotherapy alone group	(Investigator assessed, all grade) 26% (n = 41/156) sintilimab + chemotherapy vs. 16% ( n = 25/160) chemotherapy alone G5 1% (pneumonitis) vs. 1% (unknown)
Durvalumab + Osimertinib	Oxnard G et al. [44] 2022 TATTON trial Phase 1b 2nd line	23	All grade 100%. 39% discontinued due to TRAEs	Pneumonitis 22% [*n* = 2 at 3 mg/kg [*n* = 1 grade 2 *n* = 1 grade 3] *n* = 3 at 10 mg/kg [*n* = 1 grade 1, *n* = 1 grade 2 *n* = 1 grade 4] * Overall figure Not available
Nivolumab	Rizvi et al. [45] 2014 Checkmate 063 Phase 2 ≥2nd line	117	74% (n = 87) G3/4 17% (n = 20) Fatigue 33% (n = 38) Asthenia 12% (n = 14)	G5 3% Rash 11%(n = 13) Pneumonitis 3% (n = 4) Diarrhea 3% (n = 3)
Durvalumab + Osimertinib vs. Osimertinib ** Terminated early due to the TATTON trial safety concern*	Yang J et al. [46] 2019 CAURAL trial Phase 3 ≥2nd line	12	8% (n = 1) G3 rash (n = 8 [67%]) diarrhea (n = 6 [50%]) decreased appetite (n = 6 [50%])	Possible irAE 58% (n = 7) -all G1/2
Nivolumab	Mok T et al. [47] 2022 Checkmate 722 Phase 3 ≥2nd line	294	No new safety signals identified ** Specific results awaited*	* Specific results awaited
Pembrolizumab	Chih-Hsin Yang J et al. [48] 2023 KEYNOTE 789 Phase 3 ≥2nd line	492	≥3 G3 43.7% Pembrolizumab + chemotherapy vs. 38.6% chemotherapy G5 AEs 0.4% vs. 0.8%.	≥3 G3 IRAEs and infusion reactions occurred. 4.5% vs. 2.0% G5 0.4% vs. 0%
** EGFR + ALK combined **				
Durvalumab 3rd line or greater (*EGFR*+/*ALK*+)	Garassino MC et al. [49] 2018 ATLANTIC Phase 2 ≥3rd line	444 111 in cohort 1 (*EGFR*+/*ALK*+)	All grade 48% G3 4% (n = 4) G4 2% (n = 2)	AE of special interest 25% (n = 28) irAE 12% (n = 13) G3/4 2% (n = 2) Pneumonitis 2% (n = 2) * 1 G5 pneumonitis * 2 days after starting erlotinib 65days post durvalumab
Durvalumab	Naidoo et al. [35] 2022 PACIFIC subgroup analysis Phase 3 1st line -Stage III	35 (EGFR+ ALK combined) 24 (durvalumab) vs. 11 (placebo)	AEs leading to dose delays 71% vs. 18%	Radiation pneumonitis 10% vs. 4% All low grade.
** * ALK * **				
Nivolumab + Crizotinib	Spigel D et al. [50] 2018 CheckMate 370 Phase 1/2 1st line	13	62% (n = 8/13) - at least 1 ≥ G3 toxicity, 8% (n = 1) G4 pneumonitis. * Trial was halted for safety	≥G3 hepatotoxicity 38% (n = 5) 2 deaths considered -potential G5 toxicities in these patients.
Nivolumab + Ceritinib	Felip E et al. [2] 2017 Phase 1 1st or ≥2nd line	36	AEs occurring in ≥ 40% of patients: diarrhea (64%) rash (61%) ALT increase (56%) AST increase (44%) vomiting (42%) ≥G3 with ≥10% frequency ALT increase (22%) GGT increase (17%) amylase increase (11%) lipase increase (11%)	Not reported
Pembrolizumab + Crizotinib	Patel SP et al. [51] 2020 Phase 1b 1st line	9	1 G5 pneumonia (determined to be due to disease progression)	dose limiting toxicities of ≥G3 44% (n = 4/9) * prior to maximum tolerated dose being identified 33% (n = 3) AST +/- ALT rise 11% (n = 1) Fatigue 11% (n = 1) G4 pneumonitis
Atezolizumab + Alectinib	Kim DW et al. [52] 2022 Phase 1b 1st line	21	All grade 95% G3 57% Most common G3 -rash (19%) 0 G4/5 events	All grade 86% G3 43% (n = 9) G3 Rash 19% (n = 4) G3 Dyspnoea 10% (n = 2) G3 ALT increase 10% (n = 2)
Avelumab + Lorlatinib	Javelin Lung 01 [53] 2018 Phase 1b ≥2nd line	28	96.4% (n = 27) ≥G3 53.6% (n = 15)	Serious AE 39.3% (n = 11) Pneumonitis 7.1% (n = 2) AST increase, cerebral haemorrhage, confusional state, delirium and others - all 3.6% (n = 1)

Overall, evidence for the use of ICIs alone, in combination with chemotherapy, or in combination with TKIs shows a lack of efficacy in *EGFR*m NSCLC; however, the IMPOWER 150 and ORIENT-31 trial data provide a counterpoint to this, possibly due to their combination with anti-angiogenic therapy, changes to the immunogenicity of the tumor microenvironment after targeted therapy, or differences between the agents, or trial-specific anomalies. The chemotherapy/ICI combination in the IMPOWER 150 trial without bevacizumab did not have a survival benefit compared to the chemotherapy + bevacizumab arm in sensitizing *EGFR* mutations, regardless of whether they were treatment-naïve or pre-treated (all patients HR = 1.0; 95% CI: 0.57–1.74; previous TKI: HR = 1.22; 95% CI: 0.68–2.22), suggesting a possible synergistic ICI–*VEGF*-inhibitor effect [56].

It is possible that the domain involved in any oncogenic alteration may influence the efficacy and safety of ICIs, e.g., sensitizing vs. non-sensitizing *EGFR* mutations; however, given the rarity of many of these oncogenic alterations it would be extremely difficult to power trials to evaluate these differences. 

One potential confounding factor would be the positive data seen with atezolizumab and sintilimab, but overall the efficacy data are unconvincing, and this is reflected in guidelines suggesting sequential TKI and chemotherapeutic agent use rather than an ICI in *EGFR*m NSCLC [1,40,58].

### 4.2. Safety

The timing and sequencing of *EGFR* TKI in relation to ICIs is important, as there is concern regarding increased toxicity when using TKI with or following an ICI in *EGFR*m NSCLC [39,41,42,59,60]. In particular, there is concern regarding pneumonitis following treatment with *EGFR*-TKI immediately and within 3 months of ICIs [44,61]. The incidence of possible immunotherapy-related pneumonitis is 3–22% in prospective studies (Table 2). For context, in a phase 3 trial of nivolumab versus docetaxel in unselected NSCLCs, the incidence of pneumonitis in the nivolumab arm was 4% [17]. Incidences of G3/4 treatment-related adverse events (TRAEs) of up to 55% have been reported in trials combining ICI and *EGFR*-TKIs (Table 2) [39,41,42,46,59,60]. Similarly, retrospective analyses have raised concerns regarding pneumonitis in *EGFR*m patients treated with an ICI [61]. The sequencing of therapies has been postulated as a possible cause of the increased incidence of IRAEs, for example, in a retrospective review, 24% (n = 5/21) of those who started osimertinib within 90 days of a prior ICI developed severe IRAEs [59]. Conversely, 0 of 29 patients developed a severe IRAE when an ICI was given after TKI, suggesting that immunotherapy followed by *EGFR*-TKI is particularly harmful, and it is possibly safe to administer ICIs after TKI [59]. There are also a number of published case reports of severe toxicity following TKI initiation post ICIs, including a death due to toxic epidermal necrolysis 3 weeks after the initiation of osimertinib after pembrolizumab [60]. As a result of these concerns, it is common practice to withhold systemic anti-cancer treatment until NGS results are available to reduce the toxicity for patients who may harbor EGFR or other actionable oncogenic drivers, if appropriate in the clinical context. Some clinicians also initiate treatment with chemotherapy alone if it is deemed the patient should urgently start treatment while NGS results are awaited.

## 5. ALK 

The data from phase 1/2 clinical trials evaluating *ALK*-targeted therapies in combination with ICIs have demonstrated disappointing efficacy and a significant toxicity profile (Table 1 and Table 2) [2,52]. The majority of clinical studies which include *ALK* + tumors are insufficiently powered for significant subgroup analysis in the *ALK* + cohort, or *ALK* + patients are excluded altogether. In four patients in the PACIFIC trial, a mPFS of 7.8 months [95% CI, 3.9-NR] was seen in the *ALK* + patients [13]. A few small retrospective studies have demonstrated an ORR of 0–3.6%, and an mPFS of 2.3–2.5 months in *ALK* + NSCLC [12,51,62,63]. 

NCT05266846 is a single-arm phase 2 study investigating the use of pembrolizumab, bevacizumab, and chemotherapy in *ALK* + metastatic NSCLC patients who have progressed through Alectinib with persistent 5′*ALK*. 

Given its lack of efficacy and an immunosuppressive tumor microenvironment, alternative approaches may be needed to create responses to immunotherapy in *ALK* + NSCLCs [64]. Some progress has been made in developing *ALK* + cancer vaccines in mouse models of NSCLC, alone or in combination with TKI or anti PDL-1 therapy [65]. Investigation is ongoing for CAR-T cell development in *ALK* + tumors; however, this has not demonstrated clinical efficacy to date [66]. 

### Safety

Prospective data are limited regarding the side effects of combination ICI and *ALK*-directed TKIs (Table 2). The phase 1/2 CheckMate 370 trial aimed to demonstrate the safety and efficacy of nivolumab in combination with crizotinib in *ALK* + NSCLC; however, the trial was halted due to safety concerns [50]. In this trial, 5 of 13 (38%) patients developed ≥ grade 3 hepatic toxicity and there were two deaths considered potential grade 5 toxicities in these patients. Overall, 8 of 13 (62%) patients developed at least 1 ≥ grade 3 toxicity, including one grade 4 pneumonitis [50]. In a phase 1b study combining pembrolizumab and crizotinib, dose-limiting toxicities of ≥G3 were seen in four of nine patients (44%) prior to the maximum tolerated dose being identified [51].

## 6. BRAF

### 6.1. Efficacy

The efficacy of ICIs is well established in *BRAF*-mutant melanoma [67,68,69]. An ORR of 63–67%, mPFS of 9.3–14·9 m, and overall survival benefits have been demonstrated in randomized trials of targeted therapy with *BRAF/MEK* inhibitors in metastatic melanoma [70,71]. This compares to an mPFS of 16.8 m (anti-PD-1/CTLA-4), 5.6 m (anti-PD-1), and 3.4 m (anti-CTLA-4) in CHECKMATE-067, and 11.6 m (Anti-PD-1) in KEYNOTE 006, both trials of ICIs in metastatic melanoma, with a pre-specified stratification for patients with *BRAF* mutations [68,72]. 

*BRAF* mutations are less common in NSCLCs, found in 1.5–2.5%, and the ORR to *BRAF*-directed TKI therapies can be up to 64% in NSCLC [73]. We identified four published retrospective studies examining 108 patients with *BRAF*m NSCLC and their outcomes with ICIs [12,13,74,75]. In the four studies, the mPFS ranged from 1.8 to 5.3 months in the metastatic setting and was 8.4 months for stage III disease for *V600E*m patients [12,13,74,75]. For those with non-*V600E* mutations, an mPFS of 4.1–4.9 months for metastatic patients and 3.9 months for the single patient with stage III disease was estimated. A median OS of 22.5 m (95% CI 8.3-NR) in the *V600E* cohort and a mOS of 12 m (95% CI 6.8-NR) in the non-*V600E* cohort was seen in one of these studies [75]. There was no significant difference between the *BRAF V600E* and non-*V600E* mutations in terms of mPFS in the IMMUNOTARGET registry [12].

### 6.2. Uncommon Oncogenic Alterations in NSCLCs—ROS1, RET, NTRK 1/2/3, HER-2, and MET

Minimal data exist supporting the use of ICIs in many rarer oncogene-addicted subtypes such as *ROS1*, *NTRK*, and *RET*. Rare mutations were not screened for in many phase 3 trials of ICI or chemoimmunotherapy, therefore prospective evidence regarding the safety and efficacy of ICI treatment in this cohort of oncogene-addicted patients is limited.

*ROS1* fusions account for 1–3% of NSCLCs, and clinical data regarding immunotherapy use is limited mainly to case reports at present [76,77]. *ROS*1 is associated with an upregulation of PDL-1 [78]. A retrospective analysis of 28 patients with *ROS*1 alterations treated with an ICI demonstrated an ORR of 13% for those treated with a single-agent ICI and 83% for those treated with chemo–ICI combinations [77]. No difference was demonstrated between responders and non-responders in PD-L1 expression (*p* = 0.91) or TMB (*p* = 0.83). A published case report describes the partial response to chemo/immunotherapy with nivolumab (6 cycles of two-weekly dosing) followed by intracranial progression of the disease in a patient with a *ROS1* F2004L mutation after treatment with ceritinib [76]. A case series of two patients demonstrated one patient with an ongoing complete response at 2 years in the second-line setting and one ongoing stable disease/partial response at 8 months in the fourth-line setting in a non-smoking patient [79]. There is minimal prospective evidence regarding the safety of ICI therapy in *ROS*1 + NSCLC.

*RET*-rearranged NSCLC can be treated with *RET* TKIs like pralsetinib or selpercatinib, or with chemotherapy and immunotherapy, and account for 1–2% of NSCLCs [80]. In five retrospective reviews with small patient numbers, totaling 50 patients, an ORR of 0–38% to ICIs was seen [34,80,81]. No association with PDL-1 percentage was demonstrated with an mPFS of 3.4 m (95% CI, 2.1 to 5.6 months) in one of these studies [80]. 

NTRK gene fusions occur in <1% of NSCLCs [82]. It is recommended they be treated with NTRK-directed TKI where available in the second-line setting, due to results from basket trials. The pooled analysis of three trials showed an ORR of 79% to larotrectinib, an NTRK-directed TKI (95% CI 72–85), in 121 of 153 evaluable patients [83]. There is no evidence we are currently aware of regarding the efficacy or safety of ICIs in NTRK + NSCLC beyond case reports or case series, for example, a retrospective review showing an ORR of 50% (n = 1/2) [34].

In a retrospective, multicenter cohort analysis by Guisier et al., of 107 patients with rare driver mutations 26 patients (26%) experienced AEs, including 11 patients (10%) with grade 3 to grade 5 IRAEs (five colitis, two pneumonitis, and one anemia, hypophysitis, nephritis, and hepatitis) [74]. A breakdown of IRAEs by mutation was not reported. It is difficult to draw meaningful conclusions regarding these ‘other’ groups due to the heterogenous nature of their pathology, pathophysiology, and response to therapies.

Chemotherapy–ICI is standard of care in the first line for *HER*-2 + NSCLC. TKIs, ADCs, and monoclonal antibodies (mAbs) also have roles in the treatment of this subset of NSCLC, with phase 3 trials ongoing [84,85,86]. There are several types of *HER*-2 aberration in NSCLCs, ranging from *HER*-2 protein expression to *HER*-2 mutations as well as amplifications, and each may imply different outcomes to relevant targeted approaches. For example, *HER*-2 mutations occur in approximately 1–4% of NSCLCs, gene amplification in 2–5%, and protein overexpression in 2–30% [87]. Retrospective studies have shown an ORR of 7–27%, mPFS of 2.2 m (95% CI 1.7–15.2), and mOS of 20.4 m (95% CI 9.3-NR) in *HER*-2m NSCLC treated with ICIs [12,74]. This is inferior to those seen in studies investigating the use of targeted therapy as an anti-*HER*-2 directed therapy for these patients [12,84,85,86]. Conflicting evidence exists with ICI therapies in other malignancies, for example, in *HER*-2 + gastric cancer, ICI in combination with chemotherapy and trastuzumab (anti *HER*-2 mAB) in the KEYNOTE-811 phase 3 trial demonstrated an ORR of 74.4% (95% CI, 66.2–81.6) in the pembrolizumab arm and 51.9% (95% CI, 43.0–60.7) in the placebo arm [88]. G3-5 TRAEs had an incidence of 57% in both arms. The results of primary endpoints for mPFS and mOS are awaited. Despite this encouraging data in *HER*-2 + gastric cancer, in *HER*-2 + breast cancer, evidence of efficacy of ICI not encouraging. For example, no responses were seen in the phase 1b JAVELIN study with avelumab [89]. In addition, a phase 2 study of pembrolizumab added to trastuzumab in trastuzumab-refractory patients demonstrated an ORR of 15% in PDL-1 + patients and no responses were seen in PDL-1-negative patients [90].

*MET* amplification is the most common genetic alteration associated with the *MET* proto-oncogene, and it occurs in 3–4% of NSCLCs [91]. They are particularly common in sarcomatoid NSCLCs, occurring in up to 30%. *MET* + tumors are associated with high PDL1 expression. There are limited data on the safety profile of ICI treatments alone or in combination with chemotherapy or targeted therapy in *MET* + NSCLC. The standard of care treatment in *MET* + NSCLC includes crizotinib (a TKI) based on the profile 1001 study, which demonstrated an ORR of 39% and mPFS of 8 months [92]. More recently, amivantimab (a bispecific mAB of *EGFR* and *MET*) has been investigated with some promise as treatment for MET-amplified NSCLCs, particularly those with a MET exon 14 skipping mutation.

In the IMMUNOTARGET cohort, there was no significant difference between *MET* exon 14 skipping mutations and other mutations in terms of an mPFS of 3.4 m (*p* = 0.09) [12]. In a retrospective analysis of 147 patients at two cancer centers, responses were not higher in tumors with high PD-L1 expression or high TMB (tumor mutational burden), and a disappointing mPFS was seen [91]. Evidence of safety and efficacy otherwise comes from small datasets of 30 and 13 patients showing an ORR with ICIs of 36–46%, mPFS of 4.9 months (95% CI 4.6-NR), and mOS of 13.4 months (95% CI 9.4-NR) [74,93]. Four of the six respondants in this second study were non-smokers. In the 13-patient study, two grade 3 adverse events and four grade 1/2 events were documented [93]. 

### 6.3. Effect of the Immune Microenvironment and Smoking Status on ICI Response

The immune microenvironment of cancer cells plays a vital role in the effectiveness of anticancer therapies, particularly the response to an ICI (Figure 3). Some tumors have a ‘cold’ or ‘immune desert’ phenotype, with a low tumor mutational burden (TMB), relative genomic stability, low numbers of tumor-infiltrating lymphocytes (TILs), and low numbers of tertiary lymphoid structures (TLSs) [94]. Immune checkpoints such as CTLA-4, PDL-1, LAG-3, and others are often downregulated or inactivated in these tumors [94]. *EGFR*m and ALK + NSCLC are often characterized as such. It has been suggested that resistance to ICI in *EGFR-*mutated NSCLC is due to reduced TMB in the *EGFR*-mutant group compared with wildtype *EGFR,* or a ‘cold’ uninflamed tumor microenvironment which is immunosuppressive and reduced interferon gamma signature due to CD73 overexpression [95]. In addition to this, there is conflicting evidence that prior treatment with *EGFR*-directed TKI can downregulate the PDL1 expression and response to ICIs [95]. *ALK* + tumors have an immunosuppressive tumor microenvironment which activates the *PI3K-AKT* and *MEK-ERK* pathways, leading to decreased responsiveness to ICIs [94,96,97]. In contrast, treatment with EGFR-directed TKIs can reduce T cell apoptosis and increase interferon production [94]. 

On the other hand, ‘hot’ tumors are often characterized by genomic instability, high numbers of TILs, TLSs which facilitate the influx of immune cells, and a higher TMB and PDL-1 expression [98]. ‘Hot’ tumors are expected to have superior responses to ICIs compared to cold tumors. Smoking-associated cancers often have this signature, as seen in many *KRAS*m NSCLCs. *KRAS*m cancers have been shown to induce a pro-inflammatory and immunosuppressive stroma, often cooperating with other oncogenic mutations, e.g., *Rb1*, *STK11*, and *KEAP1*, to evade the host’s immune system through the induction of NF-κB and a variety of chemokines such as TNF-α and IL-6 [99]. The use of ICIs to reverse these effects can lead to improved responses in ‘hot’ tumors.

There is some evidence pointing towards the increased efficacy of ICIs in patients whose tumors harbor an oncogenic driver mutation, but who have a relevant smoking history. For example, one retrospective analysis of 186 patients treated with consolidation durvalumab for unresectable stage III NSCLC demonstrated an mPFS of 19.2 m (95% CI, 11.3-NR) in smokers with an oncogenic driver vs. 5.8 m (95% CI, 3.9-NR) in non-smokers (*p* = 0.001) [13]. This was driven to a large extent by patients with *KRAS* mutations. *KRAS*-mutant cohorts are known to be enriched for a history of smoking [23]. In the oncogene-addicted cancers in this cohort, the mPFS was not correlated with PDL-1 percentage [13].

Additionally, in the FDA pooled analysis of 1430 patients (39% *KRAS*m) from registrational clinical trials discussed above, 67% of patients were current or former smokers in the overall cohort, and 60% had a positive PDL-1 score, and no difference in outcomes was demonstrated between patients with *KRAS*m and *KRAS*wt tumours [27].

In a subgroup analysis of the IMMUNOTARGET study, the mPFS was positively associated with PD-L1 expression for patients with *KRAS* or *EGFR* mutations, and with smoking status for those with *BRAF* or *HER-2* mutations [12]. In another retrospective analysis described above, by Gainor et al., the ORR among those with ≤10 pack years was 4.2% vs. 20.6% among heavy smokers with *ALK* or *EGFR* mutations (*p* = 0.123) [63]. The ORR to ICIs in an analysis by Ng et al. was 16.9% (n = 11/65) among smokers compared to 0% (n = 0/26; [*p* = 0.019]) among never-smokers [100]. Smoking status also influenced the mPFS in this study (4.07 vs. 1.73 months; [*p* = 0.004]) [100].

Smoking status should be considered when considering ICI therapy in patients with oncogenic driver mutations, with evidence of superior responses in smokers than non-smokers across a variety of mutations. It is thought that smoking-related cancers often harbor a ‘hot’ immune microenvironment, as discussed above, which may lead to increased ICI efficacy in patients with a significant smoking history. 

### 6.4. Future Directions

We have established that ICIs may have efficacy in certain subsets of oncogene-addicted NSCLCs in the first- or second-line settings, such as *KRAS*m or *BRAF*m NSCLCs. While others, such as *EGFR*m and *ALK +* NSCLCs, sustained limited benefit from ICIs and are also at risk of potentially harmful toxicity.

Learning from the melanoma literature, the question of the optimal sequence of ICI and TKI therapies is an increasingly relevant one, and will be of relevance to patients with oncogene-addicted NSCLCs. In the phase 3 DREAMSEQ trial, a benefit in mOS was demonstrated in patients treated with an ICI prior to targeted therapy rather than vice versa in advanced *BRAF*m melanoma [101]. It will be important to study when ICI-based therapies may be best intercalated into the treatment paradigm of *KRAS*m or other subsets of NSCLCs in which there is efficacy for both targeted therapy and ICI alone, or in combination with other treatments.

In contrast to other cancers, where the combination TKI + ICI is safe and efficacious, for example when *VEGF*-directed TKIs are combined with ICIs in renal cancers, early data suggest that the combination of immunotherapy and selected TKIs in NSCLCs may have important safety concerns. Most recently, *KRAS* inhibitor + ICI combinations have demonstrated concerning safety profiles which require careful management (Table 2) [36,37]. It is possible that alternative strategies, such as sequential therapies or lead-in TKI periods, will mitigate these toxicity signals; however, definitive studies on this are awaited. Currently only *G12C* mutations are available in the clinic; however, the toxicities of newer pan-RAS inhibitors or other *KRAS* inhibitors may yield further toxicity signals [102]. 

While ICIs may not be effective for *EGFR*m or *ALK*+ NSCLCs in the metastatic setting, the effect of ICIs in earlier stages of these diseases is still being understood [12,13,35]. In the neoadjuvant setting, NEOTIDE (NCT05244213) is an ongoing phase 2 trial of sintilimab + chemotherapy in patients with activating *EGFR* mutations, and NEODANA (NCT04512430) is investigating the use of carboplatin, pemetrexed, bevacizumab, and atezolizumab in the neoadjuvant setting in *EGFR*m NSCLC. The use of these approaches rather than TKIs in the perioperative setting, guided by ADAURA, NEOADAURA, NEOS, and LAURA, will be an important focus of NSCLC research in the coming years.

## 7. Conclusions

The use of ICIs in the treatment of oncogene-addicted cancers is a challenging clinical question. The IMMUNOTARGET registry provides useful real-world guidance regarding ICI therapy in oncogene-addicted cancers, and it is important to emphasize that the benefits seen in this study were driven by *KRAS* mutations to a large extent. Like all systemic anti-cancer treatments, it is imperative to weigh the risks and benefits of ICIs with each individual patient with oncogene-addicted NSCLC. Importantly, there is a paucity of prospective data on the incidence, spectrum, and severity of IRAEs in patients treated with ICIs for oncogene-addicted tumors. Care is required when extrapolating results from retrospective studies, and further prospective studies are needed to determine the efficacy and safety of ICIs in oncogene-addicted cancers. Novel biomarkers will be crucial in determining not only who benefits from ICIs in oncogene-addicted NSCLCs, but also, crucially, which patients may be at high risk for IRAEs. 

*BRAF* and *MET*+ NSCLC patients seem to derive more benefit than some other oncogene-addicted NSCLCs from ICIs, however, this seems to be less than *KRASm* or non-oncogene-addicted NSCLCs, and treatment with additional chemotherapy may be appropriate, even in those with high PDL-1 expression, to offset this concern. This may also be the case in *KRAS*m NSCLC if *KEAP1* or *LKB1/STK11* co-mutations are present, as these seem to have a detrimental effect on *KRAS*m NSCLC patient outcomes, particularly when a single-agent ICI is used.

As noted throughout this article, difficulty arises when attempting to draw firm conclusions regarding the efficacy, clinical effectiveness, and safety of ICIs in oncogene-addicted NSCLCs. The standard of care for patients with many oncogene-addicted cancers remains TKI-directed therapies, and, indeed, for rare tumors like *RET* and *NTRK*, tumor-agnostic approvals by the FDA have been forthcoming.

It is appropriate, given the evidence at present, to consider first-line therapy in NSCLCs with mutations in *BRAF*, *MET*, or *KRAS* to be combination chemotherapy/ICI or a single-agent ICI dependent on the PDL-1 status. Indeed, many patients with *ROS-1- MET*-, *RET*-, *NTRK*-, *BRAF*-, or *HER-2*-mutated NSCLC currently receive ICIs as a first-line therapy due to the lack of genomic information, although this is reducing due to the broader use of NGS and liquid biopsy techniques. After the exhaustion of targeted treatments in *EGFR*, *ALK*, *ROS-1 MET*, *RET*, *NTRK*, and *HER-2*, ICIs may be considered as a salvage treatment, but caution must be exercised given the limited data available. 

Certainly, the use of ICIs in *EGFRm* and *ALK* + NSCLC should be carefully considered, particularly as a single-agent strategy, as the risk/benefit ratio of ICIs in these patients is debatable. If used, it should be in carefully selected patients, possibly considering the smoking exposure of the patient. The risk of immune-mediated toxicity after ICIs when these patients are treated with subsequent TKIs provides further incentive to ensure broad NGS is available prior to ICI use in NSCLC, not only to optimize therapeutic decision making, but also to minimize safety concerns in later lines of therapy.

## Figures and Tables

**Figure 1 cancers-16-00527-f001:**
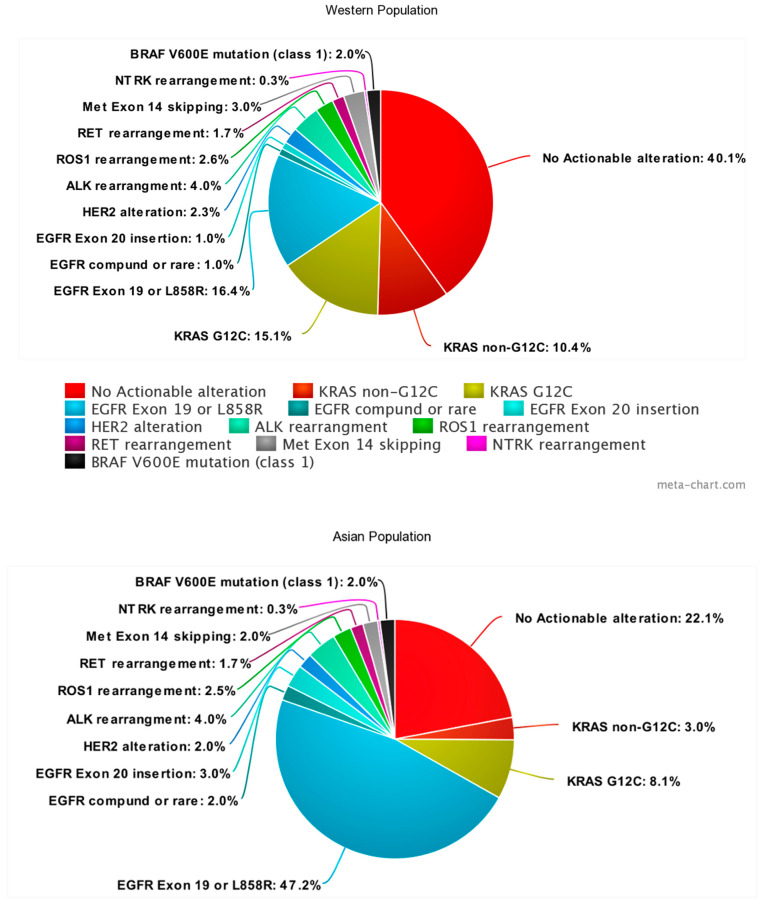
Incidence of oncogenic mutations in NSCLC. Created with meta-chart http://www.meta-chart.com/pie. accessed on 9 January 2024.

**Figure 2 cancers-16-00527-f002:**
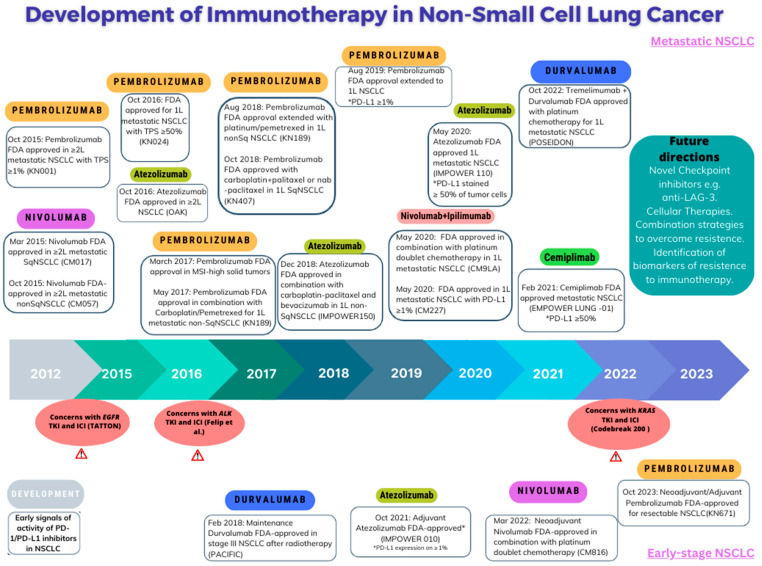
Timeline of Development of Immunotherapy and FDA Approvals for NSCLC [2].

**Figure 3 cancers-16-00527-f003:**
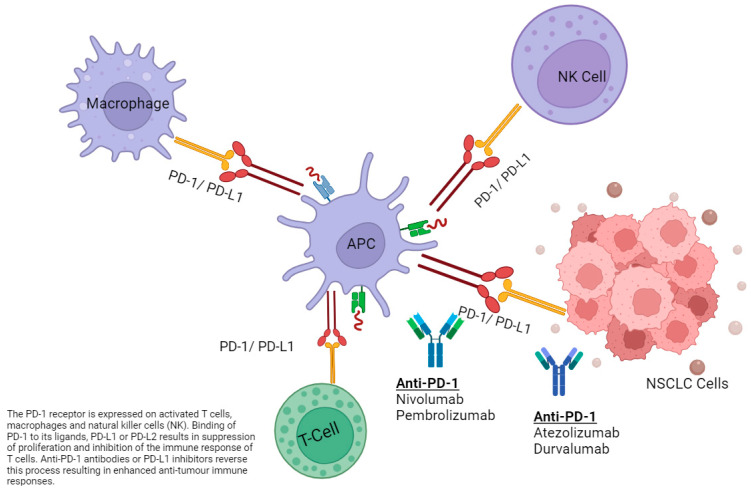
Interaction of immune checkpoints and immunotherapies in NSCLC.

## Data Availability

The data can be shared up on request.

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
