# Peer review of "Is Immunotherapy Beneficial in Patients with Oncogene-Addicted Non-Small Cell Lung Cancers? A Narrative Review"

_cancers, 2024, doi:10.3390/cancers16030527_

Round 1

Reviewer 1 Report

Comments and Suggestions for Authors

The review aims to provide a complete view of the current status of the applicability and efficiency of immunotherapy in non-small cell lung cancer. The major issues in the current format are:

1. The review is comprehensive, yet suffers from a lack of introduction . What is immunotherapy in general? what does it work, what options do exist?  Otherwise it is hard to follow for general audience.

2. Also more detail on the NSCLC mutational landscape, which ones present the so-called hot tumors; which ones are cold? what is the concept behind these definitions? the differences? which features define the status of cold vs hot tumors?

3. There is a lack of figures, which makes it hard to stay focused and interested while reading. for example a timeline about the development of immunotherapy; an overview figure about subytpes/oncogene-addictions within NSCLC.

  1.  

Reviewer 2 Report

Comments and Suggestions for Authors

The article is important as many times amongst oncologists the importance of mixing therapies (TKIs, then ICIs, or only ICSs, etc) is not clearly understood and the selection of treatment 

In the introduction, it is not  worded clearlythat the article is about NSCLC LUAD. This should be emphasized more strongly.

In subheading 2, a table should be added to make it easier to follow the text and highlight important conclusions.

Comments on the Quality of English Language

Although I do realize that the authors are likely to be native English speakers, some of the wording and the structure of some sentences are simply strange or sloppy. A good readthrough would improve the manuscript.

Author Response

Please see the attahment

Reviewer 3 Report

Comments and Suggestions for Authors

In this review article, the authors provided overview and current advance of immune checkpoint inhibitors (ICI) in non-small cell lung cancer (NSCLC) driven by single genetic alterations, summarized clinical studies of ICI, alone or in combination of with chemotherapy or targeted therapy in NSCLC. This manuscript is well written and would be an important source in understanding the use of ICI in oncogene-addicted NSCLCs.

Minor points:

Table 1 and Table 2 can be improved further with some changes below:

1.      Please make sure no typos and errors

2.      Please include references in the table

3.      Please follow the order in the text:  KRAS can be listed first, then followed by EGFR

4.      Please also list other mutations such as BRAF and some uncommon mutations described in the main text.

Round 2

Reviewer 1 Report

Comments and Suggestions for Authors

I do recognize the substantial improvement of the manuscript.

The title of the manuscript is misleading given that the main focus of the manuscript is specifically NSCLC and the use of immune checkpoint inhibitors.  Otherwise, authors would need to add other types of immunotherapies as well. In that regard, there are many reviews to consult.

Also while having the figures is nice, better structured versions would improve the quality. Figure 1 and 2 can be combined. A general overview figure of existing immunotherapies may help. Still more visual aid is needed (e.g. interaction of tumor cell with immune system).

Comments on the Quality of English Language

In the last weeks, the quality of the manuscript has been improved by adding the figures and some introduction on immune checkpoint inhibitors.  The title states the work is about immunotherapies in cancers. Yet the manuscript is specifically about NSCLC and immune checkpoint inhibitors. Either the title or the manuscript requires changes. Another criticism is about the quality of the figures 1 and 2. I am sure the authors can come up with a more aesthetic version. More figures are needed.

I recommend for publication after major revision.

Round 3

Reviewer 1 Report

Comments and Suggestions for Authors

I appreciate the changes.

Comments on the Quality of English Language

The manuscript is well-written